TOPICAL REVIEW

# Ageing and exercise-induced motor unit remodelling

Eleanor J. Jones[1], Shin-Yi Chiou[2] , Philip J. Atherton[1] , Bethan E. Phillips[1] and Mathew Piasecki[1]

[1]*Centre of Metabolism, Ageing & Physiology (COMAP), MRC–Versus Arthritis Centre of Excellence for Musculoskeletal Ageing Research, Nottingham NIHR Biomedical Research Centre, School of Medicine, University of Nottingham, Nottingham, UK*
[2]*School of Sport, Exercise, and Rehabilitation Sciences, MRC-Versus Arthritis Centre for Musculoskeletal Ageing Research, Centre for Human Brain Health, University of Birmingham, Birmingham, UK*

Edited by: Ian Forsythe & Russell Hepple

The peer review history is available in the Supporting information section of this article (https://doi.org/10.1113/JP281726#support-information-section).

**Motor unit remodelling in ageing and exercise**

Inactive
Loss of MU
Denervation without reinnervation
Muscle fibre loss

Active
Loss of MU
Denervation and reinnervation
Muscle fibre rescue

The Journal of **Physiology**

**Abstract**   A motor unit (MU) comprises the neuron cell body, its corresponding axon and each of the muscle fibres it innervates. Many studies highlight age-related reductions in the number of MUs, yet the ability of a MU to undergo remodelling and to expand to rescue denervated

 

**Eleanor Jones** is a PhD student within the Centre of Metabolism, Ageing and Physiology (COMAP), studying the effects of exercise and pharmaceutical interventions on motor unit plasticity in humans. **Mathew Piasecki** is an Assistant Professor within the same group with research interests centred around the structure and function of motor units. All authors are members of the MRC Versus Arthritis Centre for Musculoskeletal Ageing Research (CMAR), based at the Universities of Nottingham and Birmingham, UK. The aims of this research group are to apply a multi-methodological approach to generate mechanistic insight into neuromuscular decrements in human ageing and disease.

The Journal of Physiology

muscle fibres is also a defining feature of MU plasticity. Remodelling of MUs involves two coordinated processes: (i) axonal sprouting and new branching growth from adjacent surviving neurons, and (ii) the formation of key structures around the neuromuscular junction to resume muscle–nerve communication. These processes rely on neurotrophins and coordinated signalling in muscle–nerve interactions. To date, several neurotrophins have attracted focus in animal models, including brain-derived neurotrophic factor and insulin-like growth factors I and II. Exercise in older age has demonstrated benefits in multiple physiological systems including skeletal muscle, yet evidence suggests this may also extend to peripheral MU remodelling. There is, however, a lack of research in humans due to methodological limitations which are easily surmountable in animal models. To improve mechanistic insight of the effects of exercise on MU remodelling with advancing age, future research should focus on combining methodological approaches to explore the *in vivo* physiological function of the MU alongside alterations of the localised molecular environment.

(Received 17 September 2021; accepted after revision 14 February 2022; first published online 12 March 2022)

**Corresponding author** M. Piasecki: Royal Derby Hospital, Derby, UK. Email: mathew.piasecki@nottingham.ac.uk

**Abstract figure legend** Ageing is associated with a loss of motoneurons, which results in the denervation of motor unit muscle fibres and their inability to contract. Several lines of evidence from human and animal models have now highlighted the role of exercise in improving reinnervation capacity and the rescue of denervated fibres, presumably acting to preserve fibre number and total muscle function.

## Introduction

The age-related decline in muscle mass and function, known as sarcopenia, has a detrimental impact on activities of daily living and functional independence. According to the European Working Group on Sarcopenia in Older People (EWGSOP), sarcopenia is defined as 'a muscle disease (muscle failure) rooted in adverse muscle changes that accrue across a lifetime' (Cruz-Jentoft et al., 2019). Although multifactorial in origin, it can be characterised by both *atrophy* and *loss* of individual fibres, the latter being robustly associated with fibre denervation (Wilkinson et al., 2018). Motor unit (MU) remodelling compensates for the progressive reduction in the number of MUs via fibre reinnervation, which occurs through collateral axonal sprouting and new neuromuscular junction (NMJ) formation which 'rescues' adjacent denervated fibres (Gordon & Fu, 2021; Tam & Gordon, 2003; Udina et al., 2011). This process results in an increase in the size (or innervation ratio) of the MU, which may be successful up to an ill-defined point where reinnervation fails and fibre loss prevails (Piasecki et al., 2018).

The *in vivo* study of human MUs is possible via various applications of electromyography (EMG), whereby MU potentials are sampled during muscle contractions to reveal parameters representative of the MU structure and function (Del Vecchio et al., 2020; Piasecki, Garnés-Camarena et al., 2021). Applications of these methods have helped establish, at least broadly speaking, that older age is associated with a decline in MU number (McNeil et al., 2005) but an initial increase in MU size (Piasecki, Ireland, Jones et al., 2016). Denervation becomes more common with older age alongside individual fibre dysfunction, existing in a complex 'cause-or-consequence' relationship whereby it is unclear if fibre dysfunction instigates denervation, or the reverse (Anagnostou & Hepple, 2020; Hepple & Rice, 2016). If MU remodelling and fibre rescue is unsuccessful, fibre loss will be permanent (Aare et al., 2016).

Resistance exercise training (RET) is an effective intervention for sarcopenia, initiating anabolic processes leading to increased power and strength. Neural adaptations contribute to initial increases in strength in response to RET, alongside fibre changes and hypertrophy, via increases in MU discharge rate and lowering of recruitment thresholds (Del Vecchio et al., 2019), and may minimise the decrease in MU firing rate observed with advancing age (Kamen & Knight, 2004; Watanabe et al., 2018). Notably, exercise in older age may also stimulate structural neural adaptations such as the formation of new axonal sprouts (Tam & Gordon, 2003). There is mounting evidence to suggest older masters athletes (>65 years) are more successful at reinnervation of denervated fibres, theoretically creating an environment within the muscle (via chronic exercise) which enables axonal sprouting and NMJ formation. Reflecting this, in comparison to a younger group, older masters athletes exhibit electrophysiological markers of larger MU size, as well as a more homogeneous distribution of measures of motor unit potential (MUP) size and complexity across the muscle depth (Jones et al., 2021; Piasecki et al., 2019). Other

methodological approaches support this notion, with increased fibre type grouping (Zampieri et al., 2015) and fewer histological markers of denervated fibres in older athletes (Soendenbroe et al., 2021; Sonjak et al., 2019) (Fig. 1). However, the underlying mechanisms remain unclear due to the majority of the investigations on the effects of ageing and exercise on MU plasticity in humans being of cross-sectional study design. This review will appraise current evidence in humans in relation to the influence of exercise on peripheral MU remodelling in older age, and the underpinning molecular mechanisms garnered from more tractable animal models.

### Reinnervation and regrowth

**Axonal sprouting.** Human $\alpha$-motor axons are myelinated projections of a neuron responsible for conducting excitatory signals from upper motoneurons to initiate muscle contraction. The generation of motoneuron action potentials depends on the integration of synaptic inputs from descending pathways and afferent feedback from peripheral receptors. These nerves extend long distances, and utilise protein-assisted mRNA transport along axons and localised translation within axons to respond to external stimuli (Dalla Costa et al., 2021). The process of MU remodelling involves sprouting of new axonal growth cones in existing neurons, from the nodes of Ranvier, the nerve terminal or the motor endplate, and the eventual formation of new connections with denervated muscle fibres (Tam & Gordon, 2003). This can be achieved through both the formation of new NMJs (Gordon & Fu, 2021) and reinnervation of the existing postsynaptic targets (Li et al., 2018; Rantanen et al., 1995). This process is stimulated by neurotrophins in an autocrine and/or paracrine fashion, as outlined by work in animal models (English et al., 2014; Rigoni & Negro, 2020), and is known to decrease with age (Aare et al., 2016). Although much of the sprouting processes described have been generated from models of nerve damage (e.g. full and partial nerve sectioning), it can be initiated by muscle fibre inactivity

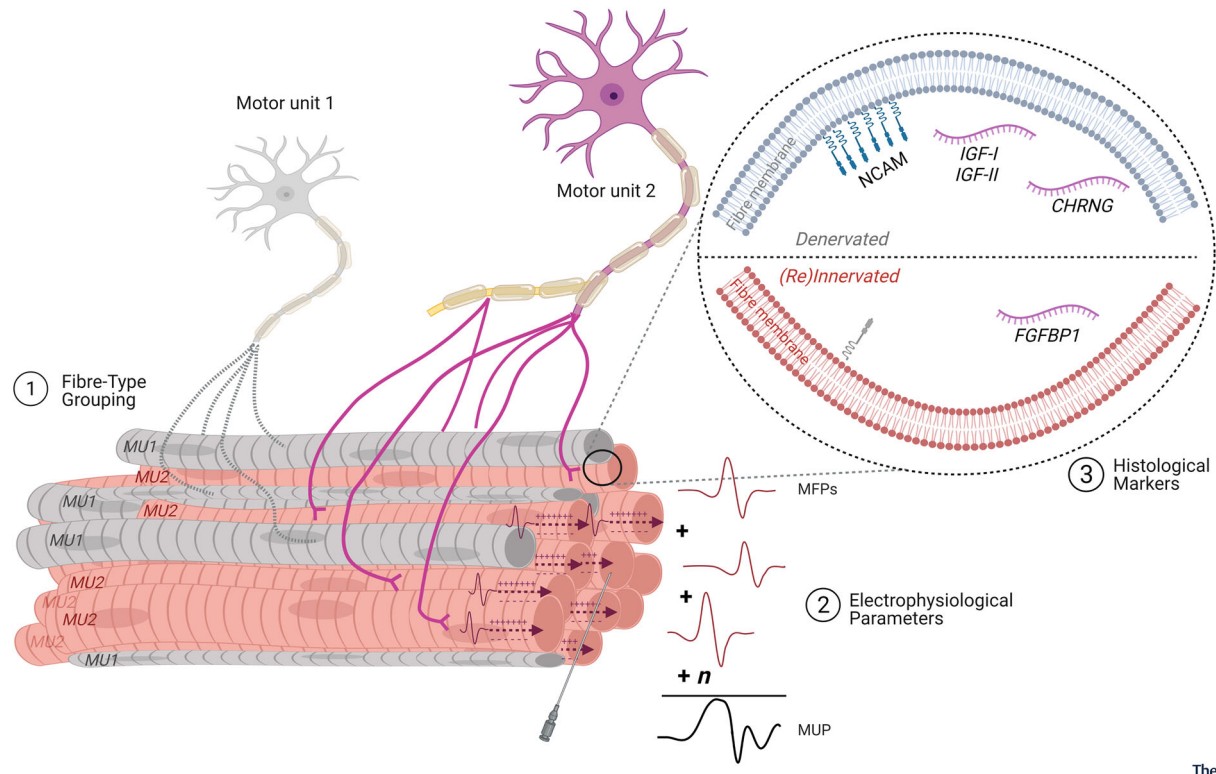

**Figure 1. Summary of human evidence to support greater reinnervation capacity in highly exercised older muscle, largely generated from studies of masters athletes**
(1) Increased fibre type grouping; fibres of MU2 are grouped, with a central fibre shown at tip of needle, entirely enclosed by fibres of the same type. (2) Larger electrophysiological markers of motor unit size; individual MFPs within recording range of a needle electrode summate to generate a MUP. (3) Fewer histological markers of denervation and altered expression of denervation-related genes. Abbreviations: CHRNG, acetylcholine $\gamma$ subunit; FGFBP1, fibroblast growth factor binding protein 1; IGF-I/II, insulin-like growth factor-I/II; MFP, muscle fibre potential; MUP, motor unit potential; NCAM, neural cell adhesion molecule.

(Tam & Gordon, 2003) – highlighting the importance of muscle–nerve interaction.

During initial axonal sprouting, the neurotrophin brain-derived neurotrophic factor (BDNF) is synthesised by motoneurons, local Schwann cells and muscle fibres (Pradhan et al., 2019). It then acts via tyrosine receptor kinase B (TrkB) signalling and p75 in neurons to upregulate production of proteins required for axonal sprouting (Pradhan et al., 2019) (Fig. 2), which may be deregulated in older age (Anisimova et al., 2020). The BDNF–TrkB signalling pathway in motor axons is well established; blocking BDNF activity and BDNF-KO mice result in reduced axon outgrowth (Wilhelm et al., 2012; Zhang et al., 2000), while TrkB agonists enhanced axon

regeneration (English et al., 2013). Circulating BDNF spatial specificity is partly controlled by calcium influx stimulating transcription of TrkB signalling components (McGregor & English, 2019), and therefore its influence in relation to axonal sprouting may be localised and have a greater effect on reinnervation in active motoneurons. BDNF levels have been reported to decrease with older age (Ziegenhorn et al., 2007), and reciprocally, to increase in response to acute exercise in older individuals (Nilsson et al., 2020).

Following experimental nerve crush, insulin-like growth factors I and II (IGF-I, IGF-II) are upregulated in nerve and muscle with both serving a number of neuroprotective roles in motoneurons (Sakowski &

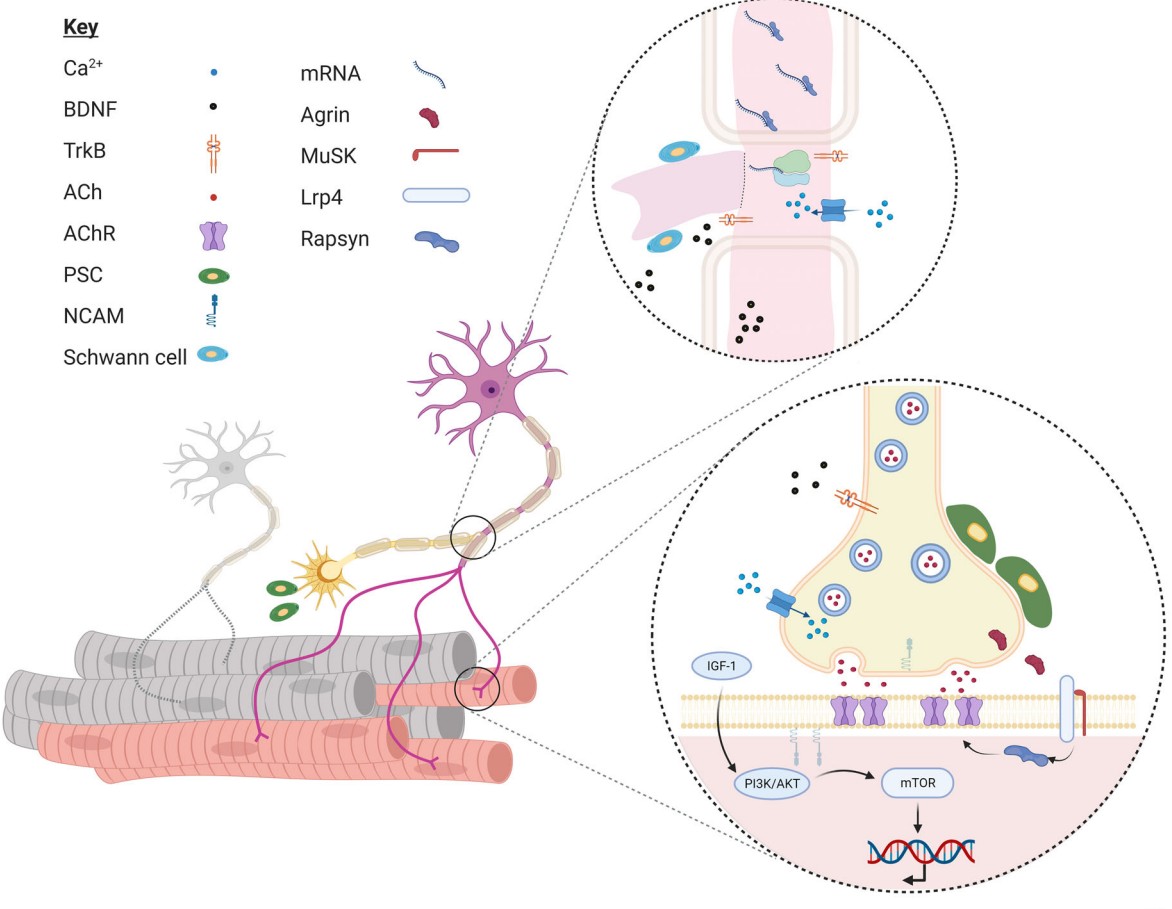

**Figure 2. The process of motor unit (MU) expansion to rescue denervated muscle fibres involves axonal branching from adjacent surviving motoneurons and the formation of new neuromuscular junctions (NMJ)**

Extending long distances, axons utilise protein-assisted mRNA transport and localised translation to alter the localised proteome to facilitate sprouting. This is mediated by a number of factors secreted by the motoneurons, muscle fibres and Schwann cells. Formation of new NMJs is largely mediated by the release of neural agrin and the Lrp4–MuSK signalling complex. Abbreviations: ACh, acetylcholine; AChR, acetylcholine receptor; BDNF, brain-derived neurotrophic factor; IGF-I, insulin-like growth factor-1; mTOR, mechanistic target of rapamycin; PI3K/AKT, phosphatidylinositol 3-kinase/protein kinase B; TrkB, tyrosine receptor kinase B; PSC, perisynaptic Schwann cell.

Feldman, 2012). IGF-I has beneficial neurotrophic and myogenic effects (Rabinovsky et al., 2003), while IGF-II is synthesised by inactive fibres in partially denervated muscles and acts as a sprouting stimulus via muscle–nerve interactions (Glazner & Ishii, 1995). Furthermore, greater expression of IGF-I and IGF-II receptors are associated with motoneurons less susceptible to degeneration in amyotrophic lateral sclerosis rodent models (Allodi et al., 2016). As a primary downstream mediator of growth hormone (GH), circulating IGF-I responds to exercise, yet outcomes from plasma measures are inconsistent, likely explained by the localised nature of its secretion and receptor binding (Birzniece, 2019). Similarly, the role of IGF and its signalling in ageing has produced contentious findings, and the evidence supporting increased life span with reduced levels is difficult to reconcile with the numerous neuroprotective effects, both peripherally and centrally (Bartke et al., 2003).

The protein kinase complex, mechanistic target of rapamycin complex 1 (mTORC1) is a key regulator of protein synthesis and is highly responsive to muscle contraction, but also plays a role in the maintenance of NMJ structure. It is clear the temporal regulation of mTORC1 following denervation is paramount, with potential mechanisms including increased synthesis of proteins required for structural NMJ stability, minimisation of mitochondrial dysfunction via maintenance of autophagic flux (Baraldo et al., 2020), and the promotion of nuclear import of denervation gene regulators (histone deacetylase HDAC4) (Castets et al., 2019). As a potent stimulator of the mTOR pathway, IGF-1 has an important role in regulating nerve regeneration through the phosphatidylinositol 3-kinase/protein kinase B (PI3K/Akt) pathway as well as promoting muscle hypertrophy (Rabinovsky, 2004). Prolonged mTOR activation appears to impair acetylcholine receptor (AChR) clustering at the NMJ thereby limiting reinnervation (Castets et al., 2019), and administration of an mTOR inhibitor reverses age-related detrimental features of the NMJ (Ham et al., 2020). Collectively these findings support the use of pharmaceutical interventions targeting the age-related imbalance of mTOR activity to reduce fibre denervation through maintenance of existing NMJ structures and/or improve reinnervation.

Perisynaptic Schwann cells (PSCs) are non-myelinated glial cells at the NMJ, sensitive to mechanical signals and synaptic transmission. As such, they may be capable of responding to denervation and chronic inactivity (Ko & Robitaille, 2015). Indeed, following denervation PSCs produce extensions to form bridges with axonal growth cones to guide sprouting to denervated fibres (Darabid et al., 2014; Tam & Gordon, 2003). Exercise is thought to encourage sprouting through promoting calcium influx into nerve terminals and upregulating glial fibrillary acidic protein, which is needed for maintaining strength and shape in the PSCs (Tam & Gordon, 2003). As a result of treadmill running, greater nerve terminal branching has been observed in plantaris and extensor digitorum longus muscles of young rats (Deschenes et al., 2016) and enhanced axon regeneration of the fibular nerve in mice following peripheral nerve injury (Sabatier et al., 2008). With chronic inactivity, often observed in older age, this calcium influx would be markedly lower and less supportive of axonal sprouting. To date, methodological limitations have prevented research determining the effect of exercise on axonal sprouting in humans, and animal models of nerve sectioning may be a poor proxy for multiple fibre denervation in aged human muscle (Jones et al., 2017).

Multiple markers of denervation and expression and transcription of certain genes are also shown to be upregulated with age (Soendenbroe et al., 2019), and some correlate with greater fibre type grouping in aged muscle and neurodegenerative disease (e.g. Parkinson's; Kelly et al., 2018). Acetylcholine receptor $\gamma$ subunit (*CHRNG*) expression is upregulated with muscle denervation in older individuals, but was shown to decrease in response to RET (Messi et al., 2016). This subunit has shown mixed results in aged muscle and may not be a reliable biomarker of denervation (Soendenbroe et al., 2021). Neural cell adhesion molecule (NCAM) is a glycoprotein expressed on the surface of neurons and muscle cells and facilitates reinnervation when upregulated, and as such has been utilised as a marker of individual fibre denervation (Messi et al., 2016). Observed decreases of NCAM$^+$ fibres were inversely proportional to increases in muscle strength following RET in older people (Messi et al., 2016), and also increased in response to unloading in healthy young muscle (Monti et al., 2021). Furthermore, in masters athletes lower levels of NCAM and upregulation of the reinnervation promoting fibroblast growth factor binding protein 1 (FGFBP1) were noted when compared to non-athletic age-matched controls (Sonjak et al., 2019) (Fig. 1). Although immunohistochemical data reveal little of MU function, collectively the data generated by these methods are supportive of exercise attenuating age-related fibre loss via increased reinnervation.

**NMJ formation.** The NMJ comprises three major components: the presynaptic axon terminal, the postsynaptic muscle fibre and the supporting PSCs. It is a specialised chemical synapse which transmits signals from motoneurons to postsynaptic nicotinic AChRs on the muscle fibre membrane to activate the release of $Ca^{2+}$ from the sarcoplasmic reticulum and stimulate contraction of sarcomeres. As such, it has been a focus of research interest in neuromuscular disorders and ageing. Like synapses found in other locations, they also demonstrate plasticity and undergo both morphological

and physiological remodelling in response to exercise, and structural degradation with age (Deschenes, 2019; Deschenes et al., 2010; Valdez et al., 2010). This can include declines in mitochondrial number and vesicles in the presynaptic terminal, and increases in postsynaptic endplate area (Jang & Van Remmen, 2011).

NMJ formation is initiated in large part by the nerve-specific isoform of agrin and muscle-specific kinase (MuSK). Neural agrin released from the motor axon activates the Lrp4–MuSK complex on the postsynaptic fibre membrane, and the sequential activation of rapsyn which clusters AChR in the post-synaptic region (Tintignac et al., 2015) (Fig. 2). During development, multiple nerve branches innervate the same fibre and, as the NMJ develops, AChR clusters mature, with retraction and synapse stabilisation occurring – stimulated by increased expression of glial cell line-derived neurotrophic factor (GDNF), which ensures a fibre is only innervated by a single motor nerve (Ham & Rüegg, 2018).

PSCs also have an important role in synaptic trans-mission and elimination during maturation, by providing structural support and phagocytically degrading redundant or degenerated axons (Alvarez-Suarez et al., 2020). Satellite cells (SC; myogenic stem cells) assist with muscle repair, remodelling and adaptation pre-dominantly of the muscle fibre (Yin et al., 2013). In young mice, a reduction in SC number contributes to impaired NMJ regeneration following denervation as a result of structural instability and declines in muscle force generating capacity (Liu et al., 2015). With age, SC numbers decrease, but increase with RET and endurance training, as measured by Pax7$^+$, possibly contributing to improving NMJ function (Kadi et al., 2004; Mackey et al., 2014; Moore et al., 2018).

The ability to histochemically image the NMJ in animal models has provided a wealth of data. In healthy young rats, RET by weighted ladder climb resulted in an expansion of the postsynaptic area (Deschenes et al., 2000, 2015), while endurance training yields remodelling in the pre- and postsynaptic areas, including greater dispersion of ACh vesicles at the presynaptic terminals and AChR in postsynaptic terminals (Deschenes et al., 2016). Research on the NMJ in aged animals has also found NMJ fragmentation (Pannerec et al., 2016), which resonates with aligned findings of MU plasticity in atrophy-resistant/susceptible muscle groups in humans (Piasecki et al., 2018). Animal NMJs also have an age-specific response to exercise, with NMJ hypertrophy occurring in the young but less so in aged animals after RET (Krause Neto et al., 2015). Similarly with endurance training, in older animals the ability of the NMJ to adapt to exercise stimuli is significantly affected in soleus and plantaris muscles in both a muscle- and fibre type-specific manner (Deschenes et al., 2011, 2016). However, fibre adaptations are still observed with increases in fibre area and alterations in fibre type composition to a greater percentage expression of type 1 fibres. Furthermore, there is variability in rat NMJ adaptations to end-urance exercise that could be age-dependent, with both cellular and subcellular components of the neuro-muscular signalling process displaying inconclusive findings (Deschenes et al., 2020); therefore further work is warranted.

Direct structural imaging of the NMJ in humans is notoriously complex and often requires full cadaveric or post-amputation limbs (Boehm et al., 2019). However, targeted biopsy techniques involving stimulation to produce a muscle twitch to locate the area of highest NMJ density have improved NMJ yield (Aubertin-Leheudre et al., 2020). Unlike animal models, in muscles obtained from humans with peripheral vascular disease, no age-related differences in human NMJ postsynaptic area were found in the lower limb – and except for a modest increase in axon diameter – the NMJ remained structurally stable with advancing age (Jones et al., 2017). Research in age-related changes in NMJs from intercostal muscles has demonstrated mixed results, with some reporting both larger and more complex postsynaptic regions (Wokke et al., 1990), while others showing no differences (Oda, 1984). However, comparisons of NMJ imaging across human studies are difficult due to different methodological considerations and muscles studied. Furthermore, data exploring additional effects of in/activity in elderly humans are scarce. Finally, the association of NMJ structure and nerve–muscle communication, or NMJ transmission instability, is not well defined.

**NMJ transmission.** Briefly, NMJ transmission refers to the release of acetylcholine (ACh) from the motoneuron and binding to the synaptic region of the muscle fibre to initiate a muscle fibre action potential and contraction. Aged rats have demonstrated a decline in NMJ transmission stability and reliability which correlated with declines in functional measures such as grip strength (Padilla et al., 2021), and voluntary running exercise improved NMJ transmission stability in older mice (Chugh et al., 2021). However, these beneficial effects of exercise on NMJ transmission are less clear in older age (Deschenes et al., 2011).

Functional NMJ transmission instability can be measured *in vivo* in humans with intramuscular EMG (iEMG) using a statistic ('jiggle') representative of the variability in consecutive MU potentials (Stålberg & Sonoo, 1994) or near fibre MU potentials (Piasecki, Garnés-Camarena et al., 2021). Longitudinal iEMG data on the effects of an exercise training intervention are unavailable; however, cross-sectional data highlight age-related increases in NMJ instability in lower limb

muscles (Hourigan et al., 2015; Piasecki, Inns et al., 2021; Piasecki, Ireland, Stashuk et al., 2016), which may reflect early stages of fibre denervation–reinnervation. In a direct comparison, older runners (80 ± 5 years) had lower NMJ instability than age-matched inactive individuals (Power et al., 2016), yet these values were still around 30−60% greater than healthy non-athletes and athletes aged 5−10 years younger from two separate studies (Hourigan et al., 2015; Piasecki, Ireland, Coulson et al., 2016).

## Conclusion and future directions

It is widely accepted that the process of peripheral MU remodelling acts as a compensatory mechanism in response to MU loss and fibre denervation to mitigate the loss of individual muscle fibres, thereby minimising total muscle loss. However, peripheral MU remodelling results in an increase in the fibre ratio: motor units become larger with respect to the number of fibres each innervates. Without any alteration to the motoneuron recruitment threshold, it is possible that increases in motoneuron fibre ratio result in a deviation from the order of MU recruitment based on MU size, which may have negative consequences on fine motor control, particularly at lower intensities. Formation of new axonal branches may result in asynchronous action potentials from individual MU fibres, observed as temporally dispersed and complex MUPs as a result of more variable axon branch lengths and/or propagation speeds. Finally, an increase in fibre innervation ratio may place increased metabolic demand on the innervating motoneuron and increase susceptibility to damage. All of these facets require human studies. Moreover, there is a lack of longitudinal research exploring these concepts, with consequently little being known about (i) the functional outcomes of MU remodelling, (ii) an identifiable time course of adaptation, and (iii) its underpinning mechanisms. Nevertheless, as the collective evidence suggests structural and functional alterations are physiologically possible, the peripheral motor system thus represents a realistic interventional target in human ageing. With MU functionality relying on fibre number, fibre size, neurotransmission and firing rates in order to sufficiently activate muscle, multiple approaches may be necessary. This may involve voluntary exercise, pharmaceutical intervention, exogenous hormone administration (Guo, Piasecki et al., 2021; Swiecicka et al., 2020) and/or electrical stimulation (Guo, Phillips et al., 2021). Finally, future mechanistic studies should aim to combine targeted biopsies to increase NMJ enrichment, omics-based approaches (Alldritt et al., 2021) and *in vivo* electrophysiological applications to explore the 'physiol-omics' of the human NMJ. Such a focus upon cellular components and signalling pathways involved in axonal sprouting could reveal targets for future therapeutics with clinical applications in populations vulnerable to muscle atrophy beyond ageing alone.

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

## Additional information

### Competing interests

The authors have no competing interests to declare.

### Author contributions

All authors contributed to the conception, design, and drafting of the work. All authors have approved the final version of

the submitted manuscript for publication and agree to be accountable for all aspects of the work in ensuring that questions related to the accuracy or integrity of any part of the work are appropriately investigated and resolved. All persons designated as authors qualify for authorship, and all those who qualify for authorship are listed.

## Funding

This work was supported by the Medical Research Council (grant number MR/P021220/1) as part of the MRC-Versus Arthritis Centre for Musculoskeletal Ageing Research awarded to the Universities of Nottingham and Birmingham, and was supported by the NIHR Nottingham Biomedical Research Centre. The views expressed are those of the author(s) and not necessarily those of the NHS, the NIHR or the Department of Health and Social Care.

## Acknowledgements

The authors thank all the research volunteers for their study participation which has contributed to aspects of this work.

## Keywords

ageing, axonal sprouting, exercise, motor unit, neuromuscular junction

## Supporting information

Additional supporting information can be found online in the Supporting Information section at the end of the HTML view of the article. Supporting information files available:

**Peer Review History**

