## [Peer Review History · The Journal of Physiology]

Ageing and exercise induced motor unit remodelling

Eleanor J Jones, Shinyi Chiou, Philip J Atherton, Bethan E Phillips, and Mathew Piasecki
DOI: 10.1113/JP281726

Corresponding author(s): Mathew Piasecki (mathew.piasecki@nottingham.ac.uk)

Review Timeline:

Submission Date:	17-Sep-2021
Editorial Decision:	05-Nov-2021
Revision Received:	20-Jan-2022
Editorial Decision:	10-Feb-2022
Revision Received:	11-Feb-2022
Accepted:	14-Feb-2022

Senior Editor: Ian Forsythe

Reviewing Editor: Russell Hepple

Transaction Report:

Dear Mathew,

Re: JP-TR-2021-281726 "Ageing and exercise induced motor unit plasticity" by Eleanor J Jones, Shinyi Chiou, Philip J Atherton, Bethan E Phillips, and Mathew Piasecki

Thank you for submitting your Topical Review to The Journal of Physiology. It has been assessed by a Reviewing Editor and by 2 expert referees and I pleased to tell you that it is considered to be acceptable for publication following satisfactory revision.

The reports are copied at the end of this email. Please address all of the points and incorporate all requested revisions, or explain in your Response to Referees why a change has not been made.

NEW POLICY: In order to improve the transparency of its peer review process The Journal of Physiology publishes online as supporting information the peer review history of all articles accepted for publication. Readers will have access to decision letters, including all Editors' comments and referee reports, for each version of the manuscript and any author responses to peer review comments. Referees can decide whether or not they wish to be named on the peer review history document.

I hope you will find the comments helpful and have no difficulty in revising your manuscript within 4 weeks.

Your revised manuscript should be submitted online using the links in Author Tasks Link Not Available. This link is to the Corresponding Author's own account, if this will cause any problems when submitting the revised version please contact us.

You should upload:

- A Word file of the complete text (including any Tables);
- An Abstract Figure, (with accompanying Legend in the article file)
- Each figure as a separate, high quality, file;
- A full Response to Referees;
- A copy of the manuscript with the changes highlighted.
- Author profile. A short biography (no more than 100 words for one author or 150 words in total for two authors) and a portrait photograph of the two leading authors on the paper. These should be uploaded, clearly labelled, with the manuscript submission. Any standard image format for the photograph is acceptable, but the resolution should be at least 300 dpi and preferably more.

- A 'Cover Art' file for consideration as the Issue's cover image;
- Appropriate Supporting Information (Video, audio or data set https://jp.msubmit.net/cgi-bin/main.plex?form_type=display_requirements#supp).

To create your 'Response to Referees' copy all the reports, including any comments from the Senior and Reviewing Editors into a Word, or similar, file and respond to each point in colour or CAPITALS. Upload this when you submit your revision.

I look forward to receiving your revised submission.

Best wishes,

Ian D. Forsythe
Deputy Editor-in-Chief
The Journal of Physiology
<https://jp.msubmit.net>
<http://jp.physoc.org>
The Physiological Society
Hodgkin Huxley House
30 Farringdon Lane
London, EC1R 3AW
UK
<http://www.physoc.org>
<http://journals.physoc.org>

EDITOR COMMENTS

Reviewing Editor:

Thank you for submitting your review article to The Journal of Physiology. Although both reviewers indicated some enthusiasm for the work, both also noted that there is considerable room to improve the quality of the writing as well as increase the depth of the analysis in several areas. Please thoroughly revise your manuscript, paying close attention to the recommendations for improving the manuscript by both reviewers.

Senior Editor:

Thanks for an interesting review. The RE and referees have some helpful suggestions; but a key issue is to improve the clarity of the writing. Use short sentences and make direct statements. Asking a trusted colleague to read through and critically comment often helps (unless all they do is to dot i's and cross t's).

While you are doing the re-write please add a diagram to illustrate your conclusions.

And also re-write the abstract to be direct - i.e. avoid phrases like "This review summarises available evidence in humans and animals..." - just state the evidence; the abstract is not a plan for an essay. Also you could hardly summarise unavailable evidence.

Here are some examples of phrases which contain empty words/phrases:

Motor unit (MU) plasticity enables compensation of the progressive reduction >> Motor unit (MU) plasticity compensates for the progressive reduction...

Reflecting this, in comparison to a younger group, older MA exhibit electrophysiological markers of larger MU size, as well as a more homogeneous distribution of MU properties across the muscle depth when compared to age-matched untrained individuals >> Older MA have larger MU size and more homogeneous properties in comparison to younger groups and age-matched untrained individuals.

As a guide to what you need to do: nearly half of your sentences are difficult to understand or have an unclear or ambiguous meaning (and some could say the same thing in half the words :-).

REFEREE COMMENTS

Referee #1:

The purpose of the topical review is to describe the evidence on the reinnervation of muscle fibres by surviving motoneurons during aging. The manuscript provides a useful update on the underlying pathways and evidence of MU remodeling observed in older adults. However, there are a number of instances in the manuscript where more specific information is needed.

Overall Comments

1. The title of the invited topical review is "Ageing and exercise induced motor unit plasticity". I suggest that the word "remodelling" is more appropriate than "plasticity", both in the title and throughout the manuscript. Plasticity has a broader meaning and could include other adaptations besides the reinnervation of abandoned muscle fibres.

2. I wonder if the section on "Reinnervation and regrowth" should be presented before the one on "Ageing effects on

peripheral MU plasticity"?

Specific Comments (Page, Line)

18 "crosstalk" may not be the most appropriate word here due to its widespread use in the EMG literature. Do you mean neurotrophic interactions between muscle and nerve?

22 neural input of muscle?

25 Define the word plasticity. Is this different from remodelling? Line 234 seems correct.

34 Provide a reference for this definition of sarcopenia.

40 Cite studies that provide data on the reinnervation mechanisms and scope.

41 Why does the reinnervation potential decline with advancing age?

50 Provide examples of the structural changes in neurons.

55 Which MU properties?

62 What is exercise plasticity?

69 McNeil et al. (2005) do not report data on MU size?

70 What is meant by a "cause-or-consequence" relationship?

72 What is MU expansion? What is the criterion for it being lacking?

75 How is fibre-type grouping evident in this figure? Is the inset meant to show membranes in denervated and reinnervated muscle fibres?

83 The generation of action potentials by motoneurons depends on the integration of synaptic inputs from descending pathways, including the brain stem, and afferent feedback from peripheral receptors.

87 Does this mean the formation of new NMJs and not connections to existing ones?

97 What is neuron excitability synaptic strength? How are neuron excitability and synaptic strength modulated?

98 Doesn't neurogenesis refer to the formation of new neurons, typically in the CNS?

101 Does the circulating BDNF elicit axon outgrowth in all muscles of an older person? If not, why not?

109 How was motoneurone resiliency measured?

128 How could a pharmaceutical intervention reduce denervation that results from the death of motoneurons?

137 Which muscles?

145 Is there any evidence on where the new NMJs are formed?

157 To activate the release of Ca²⁺ from the sarcoplasmic reticulum.

160 What are the possible structural changes?

171 What is potentiation of a nerve connection?

188 What types of fibre adaptations?

198 This section is interesting.

207 Perhaps you need a new subheading here?

213 greater instability?

236 Does reinnervation actually minimize functional loss? For sure it reduces the decline in muscle strength, but what about

other actions for which strength is not the primary determinant (e.g., manual dexterity).

239 Do you mean motoneurone size? Is there any evidence that MU remodelling contributes to an impairment of motor function?

243 What is the evidence for this suggestion? The primary determinant of force control is the common low-frequency oscillations in the discharge rates of MUs, not the variability in firing times.

245 motoneurone

Referee #2:

In the current manuscript, the authors review the current understanding of plasticity of the motor unit in response to exercise in the context of aging. This is an important topic which is relevant to the mission of the journal. Overall, the review is clearly written but lacks some depth and could be expanded and clarified in a few areas to increase impact.

Specific comments:

Abstract

1. It is not clear what the term "physiol-OMICs" is intended to mean. Maybe expand on this concept?

Introduction:

1. "The age-associated decline in muscle mass and function, known as sarcopenia, has a detrimental impact on activities of daily living and functional independence."

Sarcopenia is usually used in the context of a geriatric syndrome. All individuals lose muscle mass/strength with age, but not all develop sarcopenia. It might be helpful to clarify this point.

2. "Reflecting this, in comparison to a younger group, older MA exhibit electrophysiological markers of larger MU size, as well as a more homogeneous distribution of MU properties across the muscle depth when compared to age56 matched untrained individuals (Piasecki et al., 2019; Jones et al., 2021)."

Can motor unit size be influenced by other factors (in addition to collateral sprouting capacity and sufficiency)? Couldn't muscle fiber hypertrophy also result in these findings? Or is hypertrophy included in the definition of larger motor unit size?

3. "theoretically creating a localized environment within the muscle (via chronic exercise) which enables new NMJ formation"

Is there evidence to suggest this effect is localized? Could this also be driven by upstream processes? Axonal transport or motoneuronal functionality? The authors could look at literature investigating motor unit responses following central nervous system changes/disorders. A recent article looking at older adults and older adults with Parkinson's disease found interesting findings of motor unit plasticity following exercise that seems relevant to the current review.

<https://journals.physiology.org/doi/full/10.1152/jappphysiol.00563.2017>

Ageing effects on peripheral MU plasticity:

4. "The in-vivo study of human MUs is possible via electromyography (EMG), whereby action potentials from MU muscle fibres are sampled with regards to a number of signature parameters to generate imaging biomarkers (Del Vecchio et al., 2020; Piasecki et al., 2021b)."

This sentence is a bit long and vague. I am a little unclear what this is saying. Imaging biomarkers?

5. "Denervation becomes more common with older age in a complex "cause-or-consequence" relationship with individual fibre dysfunction (Hepple & Rice, 2016; Anagnostou & Hepple, 2020)"

Maybe expand on "cause-or-consequence" concept here? What is meant by this?

6. The authors do not describe much in regard to neuromuscular junction transmission. It might be reasonable to expand topics to include prior work in neuromuscular junction transmission (clinical and preclinical) and the effects on neuroplasticity of neurotransmission?

7. Motor unit functionality relies on number, size, neurotransmission, and firing rates in order to sufficiently activate muscle. The authors could consider expand to include more discuss of each of these areas of potential failure and plasticity? NMJ transmission and motor neuron excitability/firing rates could be expanded upon.

8. Figure 1 could probably be improved to convey the intended message

REQUIRED ITEMS:

-Please include an Abstract Figure. The Abstract Figure is a piece of artwork designed to give readers an immediate understanding of the Review Article and should summarise the main conclusions. If possible, the image should be easily 'readable' from left to right or top to bottom. It should show the physiological relevance of the Review so readers can assess the importance and content of the article. Abstract Figures should not merely recapitulate other figures in the Review. Please try to keep the diagram as simple as possible and without superfluous information that may distract from the main conclusion of the Review. Abstract Figures must be provided by authors no later than the revised manuscript stage and should be uploaded as a separate file during online submission labelled as File Type 'Abstract Figure'. Please ensure that you include the figure legend in the main article file. All Abstract Figures will be sent to a professional illustrator for redrawing and you may be asked to approve the redrawn figure before your paper is accepted.

-Please upload separate high quality figure files via the submission form.

-Author profile(s) must be uploaded via the submission form. Authors should submit a short biography (no more than 100 words for one author or 150 words in total for two authors) and a portrait photograph of the two leading authors on the paper. These should be uploaded, clearly labelled, with the manuscript submission. Any standard image format for the photograph is acceptable, but the resolution should be at least 300 dpi and preferably more. A group photograph of all authors is also acceptable, providing the biography for the whole group does not exceed 150 words.

END OF COMMENTS

JP-TR-2021-281726 "Ageing and exercise induced motor unit plasticity"

EDITOR COMMENTS

Reviewing Editor:

Thank you for submitting your review article to The Journal of Physiology. Although both reviewers indicated some enthusiasm for the work, both also noted that there is considerable room to improve the quality of the writing as well as increase the depth of the analysis in several areas. Please thoroughly revise your manuscript, paying close attention to the recommendations for improving the manuscript by both reviewers.

We are grateful to the reviewers and editor for their thorough reviewing of our paper. We have responded to individual comments below.

Senior Editor:

Thanks for an interesting review. The RE and referees have some helpful suggestions; but a key issue is to improve the clarity of the writing. Use short sentences and make direct statements. Asking a trusted colleague to read through and critically comment often helps (unless all they do is to dot i's and cross t's).

While you are doing the re-write please add a diagram to illustrate your conclusions.

And also re-write the abstract to be direct - i.e. avoid phrases like "This review summarises available evidence in humans and animals..." - just state the evidence; the abstract is not a plan for an essay. Also you could hardly summarise unavailable evidence.

Here are some examples of phrases which contain empty words/phrases:

Motor unit (MU) plasticity enables compensation of the progressive reduction >> Motor unit (MU) plasticity compensates for the progressive reduction...

Reflecting this, in comparison to a younger group, older MA exhibit electrophysiological markers of larger MU size, as well as a more homogeneous distribution of MU properties across the muscle depth when compared to age-matched untrained individuals >> Older MA have larger MU size and more homogeneous properties in comparison to younger groups and age-matched untrained individuals.

As a guide to what you need to do: nearly half of your sentences are difficult to understand or have an unclear or ambiguous meaning (and some could say the same thing in half the words :-)).

Thank you for this information. We have revised the abstract to make it more concise. We have incorporated further conclusions into Figure 1 and expanded the description as per reviewer's comments.

REFeree COMMENTS

Referee #1:

The purpose of the topical review is to describe the evidence on the reinnervation of muscle fibres by surviving motoneurons during aging. The manuscript provides a useful update on the underlying pathways and evidence of MU remodeling observed in older adults. However, there are a number of instances in the manuscript where more specific information is needed.

Thank you, we have responded to the comments below and expanded or clarified points as necessary.

Overall Comments

1. The title of the invited topical review is "Ageing and exercise induced motor unit plasticity". I suggest that the word "remodelling" is more appropriate than "plasticity", both in the title and throughout the manuscript. Plasticity has a broader meaning and could include other adaptations besides the reinnervation of abandoned muscle fibres.

We agree this is a useful modification and have amended the manuscript as suggested.

2. I wonder if the section on "Reinnervation and regrowth" should be presented before the one on "Ageing effects on peripheral MU plasticity"?

We agree with the reviewer and have implemented the most valuable parts into the introduction alongside a number of points relevant to comments below (reinnervation potential declines with age). We feel this provides a more robust introduction to the topic. These changes can be found from line 40.

Specific Comments (Page, Line)

18 "crosstalk" may not be the most appropriate word here due to its widespread use in the EMG literature. Do you mean neurotrophic interactions between muscle and nerve?

We agree that the use of crosstalk may be confusing and have changed "crosstalk" to "interactions".

22 neural input of muscle?

We have changed this sentence to:

Line 19: “Exercise in older age has demonstrated benefits in multiple physiological systems including skeletal muscle, yet evidence suggests this may also extend to peripheral MU remodelling”.

25 Define the word plasticity. Is this different from remodelling? Line 234 seems correct.

As per previous comment, we have changed this to *remodelling*. We agree this is a more useful term.

34 Provide a reference for this definition of sarcopenia.

We have expanded on the definition of sarcopenia and provided a reference for this. This now reads:

Line 30: “The age-associated decline in muscle mass and function, known as sarcopenia, has a detrimental impact on activities of daily living and functional independence. More specifically, sarcopenia is an advanced stage of decline which The European Working Group on Sarcopenia in Older People (EWGSOP) has defined as “a muscle disease (muscle failure) rooted in adverse muscle changes that accrue across a lifetime” (Cruz-Jentoft et al., 2019).”

40 Cite studies that provide data on the reinnervation mechanisms and scope.

We have included the following citations

Gordon T & Fu SY (2021). *Peripheral nerves preferentially regenerate in intramuscular endoneurial tubes to reinnervate denervated skeletal muscles. Exp Neurol*; DOI: 10.1016/j.expneurol.2021.113717.

Tam & Gordon T (2003). *Mechanisms controlling axonal sprouting at the neuromuscular junction. J Neurocytol*; DOI: 10.1023/B:NEUR.0000020635.41233.0f.

Udina, E., Cobianchi, S., Allodi, I. and Navarro, X., 2011. *Effects of activity-dependent strategies on regeneration and plasticity after peripheral nerve injuries. Annals of Anatomy-Anatomischer Anzeiger*, 193(4), pp.347-353.

41 Why does the reinnervation potential decline with advancing age?

This is a very good point, and we are not aware of any single described mechanism. However, there are several associated factors that are physiologically plausible, listed below and they have been now addressed at various points throughout the manuscript.

1. A reduced ability to support protein synthesis to enable axonal sprouting, required for axon structure and for transport (Anisimova et al., 2020; Dalla Costa et al., 2021).

2. Severely atrophied fibres in older muscle may be unable to accommodate post synaptic regions of the NMJ (Tintignac et al., 2015).

3. An age-related Schwann cell dysfunction, again impairing support for remodelling (Painter, 2017).

4. Limited neurotrophin response (Aare et al., 2016).

5. An age-related reduction in SC number, possibly impairing NMJ stability following denervation (Liu et al., 2015)

50 Provide examples of the structural changes in neurons.

This sentence now reads:

Line 57: *“Notably, exercise in older age may also stimulate structural neural adaptations such as the formation of new axons”*

55 Which MU properties?

We have added further detail and this sentence now reads:

Line 61: *“Reflecting this, in comparison to a younger group, older MA exhibit electrophysiological markers of larger MU size, as well as a more homogeneous distribution of measures of motor unit potential (MUP) size and complexity across the muscle depth when compared to age-matched untrained individuals.”*

62 What is exercise plasticity?

We have reworded this sentence to improve clarity. This now reads:

“This review will appraise current evidence in humans in relation to exercise and its influence on peripheral MU remodelling in older age, and the underpinning molecular mechanistic insight garnered – mainly from more tractable pre-clinical animal models.”

69 McNeil et al. (2005) do not report data on MU size?

The citation has been moved to support changes in MU number. This sentence now reads:

Line 43: *“Applications of these methods have helped establish, at least broadly speaking, that older age is associated with a decline in MU number (McNeil et al., 2005) but an initial increase in MU size (Piasecki et al., 2016b).”*

70 What is meant by a "cause-or-consequence" relationship?

We have expanded this section which now reads:

Line 46: *“Denervation becomes more common with older age alongside individual fibre dysfunction, existing in a complex “cause-or-consequence” relationship whereby it is unclear if fibre dysfunction instigates denervation at the NMJ, or the reverse (Hepple & Rice, 2016; Anagnostou & Hepple, 2020).”*

72 What is MU expansion? What is the criterion for it being lacking?

In this context we use the term ‘expansion’ to illustrate the increase in fibre ratio of a ‘rescuing’ MU. To simplify, we have reworded this sentence which now reads:

Line 48: *“If MU remodelling and fibre rescue is unsuccessful, fibre loss will be permanent (Aare et al., 2016; Piasecki et al., 2018).”*

75 How is fibre-type grouping evident in this figure? Is the inset meant to show membranes in denervated and reinnervated muscle fibres?

We have made several alterations to the figure, including the labelling of MU-specific muscle fibres, and expanded the description. We hope this improves clarity.

83 The generation of action potentials by motoneurons depends on the integration of

synaptic inputs form descending pathways, including the brain stem, and afferent feedback from peripheral receptors.

Thank you. We have modified this sentence which now reads:

Line 82: *“The generation of motoneuron action potentials depends on the integration of synaptic inputs from descending pathways and afferent feedback from peripheral receptors.”*

87 Does this mean the formation of new NMJs and not connections to existing ones?

The NMJ requires both pre- and post-synaptic regions to form a junction. In this instance we refer to forming new NMJs in new sites and reinnervating post-synaptic regions on the muscle fibre i.e. the site of previous NMJ. This section now reads:

Line 85: *“The process of MU remodelling involves sprouting of new axonal growth cones in existing neurons, from either the nodes of Ranvier, the nerve terminal, or the motor endplate, and the eventual formation of new connections with denervated muscle fibres (Tam & Gordon, 2003). This can be achieved through both the formation of new NMJs and reinnervation of the existing postsynaptic targets (Rantanen et al., 1995; Li et al., 2018).”*

97 What is neuron excitability synaptic strength? How are neuron excitability and synaptic strength modulated?

Thank you for highlighting this error. We have amended this section which now reads:

Line 96: *“During initial axonal sprouting, the neurotrophin brain-derived neurotrophic factor (BDNF) is synthesized by motor neurons, local Schwann cells, and muscle fibres (Pradhan et al., 2019). It then acts via tyrosine receptor kinase B (TrkB) signalling and p75 in neurons to upregulate production of proteins required for axonal sprouting (Pradhan et al., 2019) (Figure 2), which may be deregulated in older age (Anisimova et al., 2020).”*

98 Doesn't neurogenesis refer to the formation of new neurons, typically in the CNS?

Thank you, corrected as per previous comment.

101 Does the circulating BDNF elicit axon outgrowth in all muscles of an older person? If not, why not?

Although circulating levels of BDNF are not directly linked to axonal regeneration in specific muscles, the levels of BDNF decrease with age. BDNF specificity is partly controlled through calcium influx which is abundant in active muscles. We have now added a sentence to reflect this which reads:

Line 102: *“Circulating BDNF spatial specificity is partly controlled by calcium influx stimulating transcription of TrkB signalling components (McGregor & English, 2019), therefore its influence may be localised and have a greater effect on reinnervation in active muscles”.*

109 How was motoneurone resiliency measured?

We have altered this sentence to improve clarity. It now reads:

Line 111: *“Furthermore, greater expression of IGF-II and IGF-I receptors are associated with motoneurons less susceptible to degeneration in amyotrophic lateral sclerosis rodent models (Allodi et al., 2016).”*

128 How could a pharmaceutical intervention reduce denervation that results from the death of motoneurons?

Although it is not a direct causal effect, it is probable that not all denervation is a result of MN loss, particularly where instigated by fibre dysfunction. In this instance we refer to the potential of mTOR inhibitors reducing denervation and/or improving reinnervations by promoting NMJ structural integrity, as outlined in this section. We have amended this sentence to improve clarity, which now reads:

Line 130: *“Collectively these findings support the use of pharmaceutical interventions targeting the age-related imbalance of mTOR to reduce fibre denervation through maintenance of existing NMJ structures and/or improve reinnervation”*

137 Which muscles?

We have amended this sentence to include further detail. It now reads:

Line 139: *“As a result of treadmill running, greater nerve terminal branching has been observed in plantaris and EDL muscles of young rats (Deschenes et al., 2016) and enhanced axon regeneration of the fibular nerve in mice following peripheral nerve injury (Sabatier et al., 2008).”*

145 Is there any evidence on where the new NMJs are formed?

We are not aware of any evidence on the location of new NMJ formation in the context of MU remodelling. However, there is evidence from cell culture models which demonstrates reinnervation at ‘new’ post-synaptic sites (Gordon & Fu, 2021). We have added this to the manuscript:

Line 86: *“This can be achieved through both the formation of new NMJs (Gordon & Fu, 2021) and reinnervation of the existing postsynaptic targets (Rantanen et al., 1995; Li et al., 2018).”*

157 To activate the release of Ca²⁺ from the sarcoplasmic reticulum.

We have added this descriptive step. This sentence now reads:

Line 178: *“It is a specialised chemical synapse which transmits signals from motor neurons to postsynaptic nicotinic acetylcholine receptors (AChR) on the muscle fibre membrane to activate the release of Ca²⁺ from the sarcoplasmic reticulum and stimulate contraction of sarcomeres.”*

160 What are the possible structural changes?

We have amended this section which now reads:

Line 182: *“Like synapses found in other locations, they also demonstrate plasticity and undergo both morphological and physiological remodelling in response to exercise, and structural degradation with age (Valdez et al., 2010; Deschenes et al., 2010; Deschenes, 2019). This can include declines in mitochondrial number and vesicles in the presynaptic terminal and increases in postsynaptic endplate area (Jang & Van Remmen, 2011).”*

171 What is potentiation of a nerve connection?

This was an error. This section now reads:

Line 194: *“PSCs also have an important role in synaptic transmission and elimination during maturation, by providing structural support and phagocytically degrading redundant or degenerated axons (Alvarez-Suarez et al., 2020).”*

188 What types of fibre adaptations?

We have now added more detail to specify the types of fibre adaptations. This section now reads:

Line 211: *“Similarly with endurance training, although fibre adaptations are observed with increases in fibre area and alterations in fibre type composition to a greater percentage expression of type 1 fibres, in older animals the ability of the NMJ to adapt to an exercise stimuli is significantly affected in soleus and plantaris muscles in both a muscle and fibre type-specific manner (Deschenes et al., 2011, 2016).”*

198 This section is interesting.

Thank you.

207 Perhaps you need a new subheading here?

We agree this would be useful and have added the subheading “NMJ transmission”.

213 greater instability?

We have corrected this error. This sentence now reads:

Line 244: *“In a direct comparison, older runners (mean 82yrs) had lower NMJ instability than age-matched inactive individuals (Power et al., 2016), yet these values were still around 30-60% greater than healthy non-athletic, and athletic, aged 5-10 years younger from two separate studies (Hourigan et al., 2015; Piasecki et al., 2016a)”*

236 Does reinnervation actually minimize functional loss? For sure it reduces the decline in muscle strength, but what about other actions for which strength is not the primary determinant (e.g., manual dexterity).

This is an excellent point and one in which we consider regularly. An increase in MU fibre innervation ratio *may* be expected to negatively impact upon measures of fine control and/or force steadiness. We are not aware of any data to highlight this but intend to address this in future work. This sentence has been amended to read:

Line 252: *“It is widely accepted that the process of peripheral MU remodelling acts as a compensatory mechanism in response to MU loss and fibre denervation to mitigate the loss of individual muscle fibres, thereby minimising total muscle loss.”*

239 Do you mean motoneurone size? Is there any evidence that MU remodelling contributes to an impairment of motor function?

In this context we are referring to both. Similar to the point above, an increase in innervation ratio as a result of MU remodelling would not be expected to alter the recruitment threshold of the motoneuron, challenging the concept of earlier recruited MUs being smaller (depending on the extent of remodelling). However, we are not aware of any

direct evidence to highlight this as influencing functional measures such as fine motor control. Although it is conjecture, it warrants investigation and we feel it appropriate to highlight in the “*Conclusions and Future Directions*” section, but have simplified the language which now reads:

Line 256: *“Without any alteration to the motoneuron recruitment threshold it is possible that this results in a deviation from an orderly MU recruitment pattern based on MU size, which may have negative consequences on fine motor control, particularly at lower intensities”*

243 What is the evidence for this suggestion? The primary determinant of force control is the common low-frequency oscillations in the discharge rates of MUs, not the variability in firing times.

We agree with the reviewer. In this context we do not refer to increased variability of firing of MUs, but to asynchronous contraction times of fibres within a single MU. Again, as there is no evidence to suggest this impact upon functional outcomes, we have removed this part of the sentence which now reads:

Line 259: *“Formation of new axonal branches may result in asynchronous action potentials from individual MU fibres, observed as temporally dispersed and complex MUPs as a result of more variable axon branch lengths and or/propagation speeds”.*

245 motoneurone

Thank you, we have changed to motoneuron throughout the text as suggested.

Referee #2:

In the current manuscript, the authors review the current understanding of plasticity of the motor unit in response to exercise in the context of aging. This is an important topic which is relevant to the mission of the journal. Overall, the review is clearly written but lacks some depth and could be expanded and clarified in a few areas to increase impact.

Thank you for the positive comments and suggestions.

Specific comments:

Abstract

1. It is not clear what the term "physiol-OMICS" is intended to mean. Maybe expand on this concept?

This has been removed from the abstract and is included in Future Directions:

Line 269: *“Finally, future mechanistic studies should aim to combine targeted biopsies to increase NMJ enrichment, OMICS-based approaches and in-vivo electrophysiological applications to explore the “physiol-OMICS” of the human NMJ. Such focussing upon cellular components and signalling*

pathways involved in axonal regeneration could reveal targets for future therapeutics with clinical applications in populations vulnerable to muscle atrophy beyond ageing alone."

Introduction:

1. "The age-associated decline in muscle mass and function, known as sarcopenia, has a detrimental impact on activities of daily living and functional independence."

Sarcopenia is usually used in the context of a geriatric syndrome. All individuals lose muscle mass/strength with age, but not all develop sarcopenia. It might be helpful to clarify this point.

Thank you. In line with this and comments from reviewer 1 we have amended this text, which now reads:

"The age-related decline in muscle mass and function, known as sarcopenia, has a detrimental impact on activities of daily living and functional independence. According to the European Working Group on Sarcopenia in Older People (EWGSOP), sarcopenia is defined as "a muscle disease (muscle failure) rooted in adverse muscle changes that accrue across a lifetime" (Cruz-Jentoft et al., 2019)."

2. "Reflecting this, in comparison to a younger group, older MA exhibit electrophysiological markers of larger MU size, as well as a more homogeneous distribution of MU properties across the muscle depth when compared to age matched untrained individuals (Piasecki et al., 2019; Jones et al., 2021)."

Can motor unit size be influenced by other factors (in addition to collateral sprouting capacity and sufficiency)? Couldn't muscle fiber hypertrophy also result in these findings? Or is hypertrophy included in the definition of larger motor unit size?

The size (area, amplitude, duration) of a motor unit potential (MUP) is influenced by the number of fibres, and the combined circumference of each fibre, contributing to the potential. It is also largely influenced by the distance between the recording electrode and the signal origin, making surface-based measures less reliable than intramuscular. It is possible that muscle hypertrophy would result in larger MUPs, yet we find no relationship between muscle size and MUP size. Furthermore, it is typical for older masters athletes to have muscles of a similar size to age-matched non-athletes, particularly in older endurance athletes (Sipila & Suominen, 1991; Piasecki et al., 2019). Simulation studies of MU remodelling also support the increases intramuscularly recorded MUP area as a result of increase innervation ratio (greater fibre number) (Piasecki et al., 2021). These findings are also supported by human biopsy studies of masters athletes (Mosole et al., 2014; Zampieri et al., 2015).

3. "theoretically creating a localized environment within the muscle (via chronic exercise) which enables new NMJ formation"

Is there evidence to suggest this effect is localized? Could this also be driven by upstream processes? Axonal transport or motoneuronal functionality? The authors could look at literature investigating motor unit responses following central nervous system changes/disorders. A recent article looking at older adults and older adults with Parkinson's

disease found interesting findings of motor unit plasticity following exercise that seems relevant to the current review.

<https://journals.physiology.org/doi/full/10.1152/jappphysiol.00563.2017>

Thank you for the suggestions, and we agree the above reference is relevant to this review and have incorporated it. We intended this sentence to specifically refer to axonal sprouting and NMJ formation. There is strong evidence to suggest depolarisation-induced calcium influx is required to support this (discussed in review), which does hint towards localised effects i.e. when comparing active vs inactive MUs/muscles. Nevertheless, it is not entirely clear, therefore we have amended this sentence which now reads:

Line 58: *“There is mounting evidence to suggest older master’s athletes (MA) (>65yrs) are more successful at reinnervation of denervated fibres, theoretically creating an environment within the muscle (via chronic exercise) which enables axonal sprouting and NMJ formation.”*

Ageing effects on peripheral MU plasticity:

4. "The in-vivo study of human MUs is possible via electromyography (EMG), whereby action potentials from MU muscle fibres are sampled with regards to a number of signature parameters to generate imaging biomarkers (Del Vecchio et al., 2020; Piasecki et al., 2021b)."

This sentence is a bit long and vague. I am a little unclear what this is saying. Imaging biomarkers?

We agree this is overly complex and have amended this sentence, which now reads:

Line 41: *“The in-vivo study of human MUs is possible via various applications of electromyography (EMG), whereby MU potentials are sampled during muscle contractions to reveal parameters representative of the MU structure and function (Del Vecchio et al., 2020; Piasecki et al., 2021)”*

5. "Denervation becomes more common with older age in a complex "cause-or-consequence" relationship with individual fibre dysfunction (Hepple & Rice, 2016; Anagnostou & Hepple, 2020)"

Maybe expand on "cause-or-consequence" concept here? What is meant by this?

Also noted by reviewer 1. We have amended this section to improve clarity. It now reads:

Line 46: *“Denervation becomes more common with older age alongside individual fibre dysfunction, existing in a complex “cause-or-consequence” relationship whereby it is unclear if fibre dysfunction instigates denervation at the NMJ, or the reverse.”*

6. The authors do not describe much in regard to neuromuscular junction transmission. It might be reasonable to expand topics to include prior work in neuromuscular junction

transmission (clinical and preclinical) and the effects on neuroplasticity of neurotransmission?

We have added brief details on NMJ transmission and highlighted recent relevant findings from animal models. This section is as follows:

Line 227: 2.3 NMJ transmission

“Briefly, NMJ transmission refers to the release of acetylcholine (ACh) from the motoneuron and binding to the synaptic region of the muscle fibre to initiate a muscle fibre action potential and contraction. Aged rats have demonstrated a decline in NMJ transmission stability and reliability which correlated with declines in functional measures such as grip strength (Padilla et al., 2021), and voluntary running exercise improved NMJ transmission stability in older mice (Chugh et al., 2021). However, these beneficial effects of exercise on NMJ transmission are certainly suppressed in older age (Deschenes et al., 2011).”

7. Motor unit functionality relies on number, size, neurotransmission, and firing rates in order to sufficiently activate muscle. The authors could consider expand to include more discuss of each of these areas of potential failure and plasticity? NMJ transmission and motor neuron excitability/firing rates could be expanded upon.

We agree these are important aspects of motor unit function but feel an in-depth analysis of each of these may be beyond the scope of this review and will drastically increase the word count. We have briefly expanded upon NMJ transmission from lines 228.

We have also added further detail on MU FR in response to ageing and exercise from lines 56. However, again we have kept this brief as largely centrally mediated adaptations (e.g. FR) would likely require a stand-alone review.

8. Figure 1 could probably be improved to convey the intended message

Thank you, we agree. We have made several modifications to the figure, including increasing within-figure labelling, and have expanded the description in the figure legend.

References

Aare S, Spendiff S, Vuda M, Elkrief D, Perez A, Wu Q, Mayaki D, Hussain SNA, Hettwer S & Hepple RT (2016). Failed reinnervation in aging skeletal muscle. *Skelet Muscle*; DOI: 10.1186/s13395-016-0101-y.

Alldritt I, Greenhaff PL & Wilkinson DJ (2021). Metabolomics as an Important Tool for Determining the Mechanisms of Human Skeletal Muscle Deconditioning. *Int J Mol Sci* 2021, Vol 22, Page 13575 **22**, 13575.

Allodi I, Comley L, Nichterwitz S, Nizzardo M, Simone C, Benitez JA, Cao M, Corti S & Hedlund E (2016). Differential neuronal vulnerability identifies IGF-2 as a protective factor in ALS. *Sci Rep*; DOI: 10.1038/srep25960.

Alvarez-Suarez P, Gawor M & Prószyński TJ (2020). Perisynaptic schwann cells - The multitasking cells at the developing neuromuscular junctions. *Semin Cell Dev Biol* **104**, 31–38.

- Anagnostou M-E & Hepple RT (2020). Mitochondrial Mechanisms of Neuromuscular Junction Degeneration with Aging. *Cells* **9**, 197.
- Anisimova AS, Meerson MB, Gerashchenko M V., Kulakovskiy I V., Dmitriev SE & Gladyshev VN (2020). Multifaceted deregulation of gene expression and protein synthesis with age. *Proc Natl Acad Sci U S A*; DOI: 10.1073/pnas.2001788117.
- Chugh D, Iyer CC, Bobbili P, Blatnik AJ, Kaspar BK, Meyer K, Burghes AH, Clark BC & Arnold WD (2021). Voluntary wheel running with and without follistatin overexpression improves NMJ transmission but not motor unit loss in late life of C57BL/6J mice. *Neurobiol Aging* **101**, 285–296.
- Cruz-Jentoft AJ et al. (2019). Sarcopenia: Revised European consensus on definition and diagnosis. *Age Ageing*; DOI: 10.1093/ageing/afy169.
- Dalla Costa I, Buchanan CN, Zdradzinski MD, Sahoo PK, Smith TP, Thames E, Kar AN & Twiss JL (2021). The functional organization of axonal mRNA transport and translation. *Nat Rev Neurosci* **22**, 77–91.
- Deschenes MR (2019). Adaptations of the neuromuscular junction to exercise training. *Curr Opin Physiol* **10**, 10–16.
- Deschenes MR, Kressin KA, Garratt RN, Leathrum CM & Shaffrey EC (2016). Effects of exercise training on neuromuscular junction morphology and pre- to post-synaptic coupling in young and aged rats. *Neuroscience* **316**, 167–177.
- Deschenes MR, Roby MA, Eason MK & Harris MB (2010). Remodeling of the neuromuscular junction precedes sarcopenia related alterations in myofibers. *Exp Gerontol* **45**, 389–393.
- Deschenes MR, Roby MA & Glass EK (2011). Aging influences adaptations of the neuromuscular junction to endurance training. *Neuroscience* **190**, 56–66.
- Gordon T & Fu SY (2021). Peripheral nerves preferentially regenerate in intramuscular endoneurial tubes to reinnervate denervated skeletal muscles. *Exp Neurol*; DOI: 10.1016/j.expneurol.2021.113717.
- Hepple RT & Rice CL (2016). Innervation and neuromuscular control in ageing skeletal muscle. *Authors J Physiol C* **594**, 1965–1978.
- Hourigan ML, McKinnon NB, Johnson M, Rice CL, Stashuk DW & Doherty TJ (2015). Increased motor unit potential shape variability across consecutive motor unit discharges in the tibialis anterior and vastus medialis muscles of healthy older subjects. *Clin Neurophysiol* **126**, 2381–2389.
- Jang YC & Van Remmen H (2011). Age-associated alterations of the neuromuscular junction. *Exp Gerontol* **46**, 193–198.
- Li L, Xiong WC & Mei L (2018). Neuromuscular Junction Formation, Aging, and Disorders. <https://doi.org/10.1146/annurev-physiol-022516-034255> **80**, 159–188.
- Liu W, Wei-LaPierre L, Klose A, Dirksen RT & Chakkalakal J V. (2015). Inducible depletion of adult skeletal muscle stem cells impairs the regeneration of neuromuscular junctions. *Elife*; DOI: 10.7554/eLife.09221.
- McGregor CE & English AW (2019). The role of BDNF in peripheral nerve regeneration: Activity-dependent treatments and Val66Met. *Front Cell Neurosci* **12**, 522.

- McNeil CJ, Doherty TJ, Stashuk DW & Rice CL (2005). Motor unit number estimates in the tibialis anterior muscle of young, old, and very old men. *Muscle Nerve* **31**, 461–467.
- Mosole S et al. (2014). Long-Term High-Level Exercise Promotes Muscle Reinnervation With Age. *J Neuropathol Exp Neurol* **73**, 284–294.
- Padilla CJ, Harrigan ME, Harris H, Schwab JM, Rutkove SB, Rich MM, Clark BC & Arnold WD (2021). Profiling age-related muscle weakness and wasting: neuromuscular junction transmission as a driver of age-related physical decline. *GeroScience* **43**, 1265–1281.
- Painter MW (2017). Aging Schwann cells: mechanisms, implications, future directions. *Curr Opin Neurobiol*; DOI: 10.1016/j.conb.2017.10.022.
- Piasecki M, Garnés-Camarena O & Stashuk DW (2021). Near-Fiber Electromyography. *Clin Neurophysiol*; DOI: <https://doi.org/10.1016/j.clinph.2021.02.008>.
- Piasecki M, Ireland A, Coulson J, Stashuk DW, Hamilton-Wright A, Swiecicka A, Rutter MK, McPhee JS & Jones DA (2016a). Motor unit number estimates and neuromuscular transmission in the tibialis anterior of master athletes: evidence that athletic older people are not spared from age-related motor unit remodeling. *Physiol Rep*; DOI: 10.14814/phy2.12987.
- Piasecki M, Ireland A, Jones DA & McPhee JS (2016b). Age-dependent motor unit remodelling in human limb muscles. *Biogerontology* **17**, 485–496.
- Piasecki M, Ireland A, Piasecki J, Degens H, Stashuk DW, Swiecicka A, Rutter MK, Jones DA & McPhee JS (2019). Long-term endurance and power training may facilitate motor unit size expansion to compensate for declining motor unit numbers in older age. *Front Physiol*; DOI: 10.3389/fphys.2019.00449.
- Piasecki M, Ireland A, Piasecki J, Stashuk DW, Swiecicka A, Rutter MK, Jones DA & McPhee JS (2018). Failure to expand the motor unit size to compensate for declining motor unit numbers distinguishes sarcopenic from non-sarcopenic older men. *J Physiol* **596**, 1627–1637.
- Power GA, Allen MD, Gilmore KJ, Stashuk DW, Doherty TJ, Hepple RT, Taivassalo T & Rice CL (2016). Motor unit number and transmission stability in octogenarian world class athletes: Can age-related deficits be outrun? *J Appl Physiol* **121**, 1013–1020.
- Pradhan J, Noakes PG & Bellingham MC (2019). The Role of Altered BDNF/TrkB Signaling in Amyotrophic Lateral Sclerosis. *Front Cell Neurosci* **0**, 368.
- Rantanen J, Ranne J, Hurme T & Kalimo H (1995). Denervated Segments of Injured Skeletal Muscle Fibers Are Reinnervated by Newly Formed Neuromuscular Junctions. *J Neuropathol Exp Neurol* **54**, 188–194.
- Sabatier MJ, Redmon N, Schwartz G & English AW (2008). Treadmill training promotes axon regeneration in injured peripheral nerves. *Exp Neurol*; DOI: 10.1016/j.expneurol.2008.02.013.
- Sipila S & Suominen H (1991). Ultrasound imaging of the quadriceps muscle in elderly athletes and untrained men. *Muscle Nerve*; DOI: 10.1002/mus.880140607.
- Siu LT & Gordon T (2003). Mechanisms controlling axonal sprouting at the neuromuscular junction. *J Neurocytol*; DOI: 10.1023/B:NEUR.0000020635.41233.0f.
- Tintignac LA, Brenner H-RR, Ruegg MA & Rüegg MA (2015). Mechanisms Regulating

Neuromuscular Junction Development and Function and Causes of Muscle Wasting. *Physiol Rev* **95**, 809–852.

Valdez G, Tapia JC, Kang H, Clemenson Jr. GD, Gage FH, Lichtman JW & Sanes JR (2010). Attenuation of age-related changes in mouse neuromuscular synapses by caloric restriction and exercise. *Proc Natl Acad Sci U S A* **107**, 14863–14868.

Del Vecchio A, Holobar A, Falla D, Felici F, Enoka RM & Farina D (2020). Tutorial: Analysis of motor unit discharge characteristics from high-density surface EMG signals. *J Electromyogr Kinesiol* **53**, 102426.

Zampieri S et al. (2015). Lifelong physical exercise delays age-associated skeletal muscle decline. *J Gerontol A Biol Sci Med Sci* **70**, 163–173.

Dear Mathew,

Re: JP-TR-2022-281726R1 "Ageing and exercise induced motor unit remodelling" by Eleanor J Jones, Shinyi Chiou, Philip J Atherton, Bethan E Phillips, and Mathew Piasecki

Thank you for submitting your Topical Review to The Journal of Physiology. It has been assessed by a Reviewing Editor and by 2 expert referees and I am pleased to tell you that it is considered to be acceptable for publication following satisfactory revision.

The reports are copied at the end of this email. Please address all of the points and incorporate all requested revisions, or explain in your Response to Referees why a change has not been made.

NEW POLICY: In order to improve the transparency of its peer review process The Journal of Physiology publishes online as supporting information the peer review history of all articles accepted for publication. Readers will have access to decision letters, including all Editors' comments and referee reports, for each version of the manuscript and any author responses to peer review comments. Referees can decide whether or not they wish to be named on the peer review history document.

I hope you will find the comments helpful and have no difficulty in revising your manuscript within 4 weeks.

Your revised manuscript should be submitted online using the links in Author Tasks Link Not Available. This link is to the Corresponding Author's own account, if this will cause any problems when submitting the revised version please contact us.

You should upload:

- A Word file of the complete text (including any Tables);
- An Abstract Figure, (with accompanying Legend in the article file)
- Each figure as a separate, high quality, file;
- A full Response to Referees;
- A copy of the manuscript with the changes highlighted.
- Author profile. A short biography (no more than 100 words for one author or 150 words in total for two authors) and a portrait photograph of the two leading authors on the paper. These should be uploaded, clearly labelled, with the manuscript submission. Any standard image format for the photograph is acceptable, but the resolution should be at least 300 dpi and preferably more.

- A 'Cover Art' file for consideration as the Issue's cover image;
- Appropriate Supporting Information (Video, audio or data set https://jp.msubmit.net/cgi-bin/main.plex?form_type=display_requirements#supp).

To create your 'Response to Referees' copy all the reports, including any comments from the Senior and Reviewing Editors into a Word, or similar, file and respond to each point in colour or CAPITALS. Upload this when you submit your revision.

I look forward to receiving your revised submission.

Best wishes

Ian

Ian D. Forsythe
Deputy Editor-in-Chief
The Journal of Physiology
<https://jp.msubmit.net>
<http://jp.physoc.org>
The Physiological Society
Hodgkin Huxley House
30 Farringdon Lane
London, EC1R 3AW
UK
<http://www.physoc.org>
<http://journals.physoc.org>

EDITOR COMMENTS

Reviewing Editor:

Thank you for submitting your work to The Journal of Physiology. Both reviewers have indicated that the revisions have improved the manuscript. There remain a few minor points from reviewer 2 that will need to be addressed.

REFeree COMMENTS

Referee #1:

The revised manuscript is much better and I have no additional comments.

Referee #2:

Overall, the authors have addressed the feedback.

1. There is one sentence that seems unclear or possibly incorrect.

"Notably, exercise in older age may stimulate structural neural adaptations such as the formation of new axons (Siu & Gordon, 2003)."

2. I think the authors mean sprouting of axons (or axon sprouts) rather than new axons?

Also, I think this reference (first author name) is incorrect in the reference list.

3. Recommend editing for clarity and succinctness. There are several sentences that should be shortened and clarified in the abstract and main text. For example the last sentence of the abstract is too long and unclear.

REQUIRED ITEMS:

-Your MS must include a complete "Additional information section" with the following 4 headings and content:

Competing Interests: A statement regarding competing interests. If there are no competing interests, a statement to this effect must be included. All authors should disclose any conflict of interest in accordance with journal policy.

Author contributions: Each author should take responsibility for a particular section of the study and have contributed to writing the paper. Acquisition of funding, administrative support or the collection of data alone does not justify authorship; these contributions to the study should be listed in the Acknowledgements. Additional information such as 'X and Y have contributed equally to this work' may be added as a footnote on the title page.

It must be stated that all authors approved the final version of the manuscript and that all persons designated as authors qualify for authorship, and all those who qualify for authorship are listed.

Funding: Authors must indicate all sources of funding, including grant numbers. If authors have not received funding, this must be stated.

It is the responsibility of authors funded by RCUK to adhere to their policy regarding funding sources and underlying research material. The policy requires funding information to be included within the acknowledgement section of a paper. Guidance on how to acknowledge funding information is provided by the Research Information Network. The policy also

requires all research papers, if applicable, to include a statement on how any underlying research materials, such as data, samples or models, can be accessed. However, the policy does not require that the data must be made open. If there are considered to be good or compelling reasons to protect access to the data, for example commercial confidentiality or legitimate sensitivities around data derived from potentially identifiable human participants, these should be included in the statement.

Acknowledgements: Acknowledgements should be the minimum consistent with courtesy. The wording of acknowledgements of scientific assistance or advice must have been seen and approved by the persons concerned. This section should not include details of funding.

END OF COMMENTS

1st Confidential Review

20-Jan-2022

JP-TR-2021-281726 "Ageing and exercise induced motor unit plasticity"

EDITOR COMMENTS

Reviewing Editor:

Thank you for submitting your work to The Journal of Physiology. Both reviewers have indicated that the revisions have improved the manuscript. There remain a few minor points from reviewer 2 that will need to be addressed.

REFEREE COMMENTS

Referee #1:

The revised manuscript is much better and I have no additional comments.

Thank you for taking the time to re-review our manuscript.

Referee #2:

Overall, the authors have addressed the feedback.

1. There is one sentence that seems unclear or possibly incorrect.

"Notably, exercise in older age may stimulate structural neural adaptations such as the formation of new axons (Siu & Gordon, 2003)."

2. I think the authors mean sprouting of axons (or axon sprouts) rather than new axons?

Thank you. We have amended this sentence to improve clarity, which now reads:

Notably, exercise in older age may stimulate structural neural adaptations such as the formation of new axonal sprouts (Tam & Gordon, 2003).

Also, I think this reference (first author name) is incorrect in the reference list.

Thank you. We have corrected this reference.

3. Recommend editing for clarity and succinctness. There are several sentences that should be shortened and clarified in the abstract and main text. For example the last sentence of the abstract is too long and unclear.

We have made several further edits to the manuscript in order to improve clarity.

Dear Mathew,

Re: JP-TR-2022-281726R2 "Ageing and exercise induced motor unit remodelling" by Eleanor J Jones, Shinyi Chiou, Philip J Atherton, Bethan E Phillips, and Mathew Piasecki

I am pleased to tell you that your Topical Review article has been accepted for publication in The Journal of Physiology, subject to any modifications to the text that may be required by the Journal Office to conform to House rules.

NEW POLICY: In order to improve the transparency of its peer review process The Journal of Physiology publishes online as supporting information the peer review history of all articles accepted for publication. Readers will have access to decision letters, including all Editors' comments and referee reports, for each version of the manuscript and any author responses to peer review comments. Referees can decide whether or not they wish to be named on the peer review history document.

The last Word version of the paper submitted will be used by the Production Editors to prepare your proof. When this is ready you will receive an email containing a link to Wiley's Online Proofing System. The proof should be checked and corrected as quickly as possible.

All queries at proof stage should be sent to tjp@wiley.com

The accepted version of the manuscript will be published online, prior to copy editing in the Accepted Articles section.

Are you on Twitter? Once your paper is online, why not share your achievement with your followers. Please tag The Journal (@jphysiol) in any tweets and we will share your accepted paper with our 22,000+ followers!

Best wishes,

Ian

Ian D. Forsythe
Deputy Editor-in-Chief
The Journal of Physiology
<https://jp.msubmit.net>
<http://jp.physoc.org>
The Physiological Society
Hodgkin Huxley House
30 Farringdon Lane
London, EC1R 3AW
UK
<http://www.physoc.org>
<http://journals.physoc.org>

*** IMPORTANT NOTICE ABOUT OPEN ACCESS ***

Information about Open Access policies can be found here <https://physoc.onlinelibrary.wiley.com/hub/access-policies>

To assist authors whose funding agencies mandate public access to published research findings sooner than 12 months after publication The Journal of Physiology allows authors to pay an open access (OA) fee to have their papers made freely available immediately on publication.

You will receive an email from Wiley with details on how to register or log-in to Wiley Authors Services where you will be able to place an OnlineOpen order.

You can check if your funder or institution has a Wiley Open Access Account here <https://authorservices.wiley.com/author-resources/Journal-Authors/licensing-and-open-access/open-access/author-compliance-tool.html>

Your article will be made Open Access upon publication, or as soon as payment is received.

If you wish to put your paper on an OA website such as PMC or UKPMC or your institutional repository within 12 months of publication you must pay the open access fee, which covers the cost of publication.

OnlineOpen articles are deposited in PubMed Central (PMC) and PMC mirror sites. Authors of OnlineOpen articles are permitted to post the final, published PDF of their article on a website, institutional repository, or other free public server, immediately on publication.

Note to NIH-funded authors: The Journal of Physiology is published on PMC 12 months after publication, NIH-funded

authors DO NOT NEED to pay to publish and DO NOT NEED to post their accepted papers on PMC.

EDITOR COMMENTS

Reviewing Editor:

Thank you for submitting your revised work to The Journal of Physiology. All prior concerns have now been satisfactorily addressed.

Senior Editor:

Thank you for an interesting Review. I look forward to seeing it in Press.

2nd Confidential Review

11-Feb-2022